# Biochemical Characterization of Parsley Glycosyltransferases Involved in the Biosynthesis of a Flavonoid Glycoside, Apiin

**DOI:** 10.3390/ijms242317118

**Published:** 2023-12-04

**Authors:** Song An, Maho Yamashita, Sho Iguchi, Taketo Kihara, Eri Kamon, Kazuya Ishikawa, Masaru Kobayashi, Takeshi Ishimizu

**Affiliations:** 1College of Life Sciences, Ritsumeikan University, 1-1-1 Nojihigashi, Kusatsu 525-8577, Shiga, Japan; 2Ritsumeikan Global Innovation Research Organization, Ritsumeikan University, 1-1-1 Nojihigashi, Kusatsu 525-8577, Shiga, Japan; 3Graduate School of Agriculture, Kyoto University, Kyoto 606-8502, Kyoto, Japan

**Keywords:** apiin, flavonoid glycoside, glycosyltransferase, parsley, specialized metabolite

## Abstract

The flavonoid glycoside apiin (apigenin 7-*O*-[β-D-apiosyl-(1→2)-β-D-glucoside]) is abundant in apiaceous and asteraceous plants, including celery and parsley. Although several enzymes involved in apiin biosynthesis have been identified in celery, many of the enzymes in parsley (*Petroselinum crispum*) have not been identified. In this study, we identified parsley genes encoding the glucosyltransferase, PcGlcT, and the apiosyltransferase, PcApiT, that catalyze the glycosylation steps of apiin biosynthesis. Their substrate specificities showed that they were involved in the biosynthesis of some flavonoid 7-*O*-apiosylglucosides, including apiin. The expression profiles of *PcGlcT* and *PcApiT* were closely correlated with the accumulation of flavonoid 7-*O*-apiosylglucosides in parsley organs and developmental stages. These findings support the idea that *PcGlcT* and *PcApiT* are involved in the biosynthesis of flavonoid 7-*O*-apiosylglucosides in parsley. The identification of these genes will elucidate the physiological significance of apiin and the development of apiin production methods.

## 1. Introduction

Parsley (*Petroselinum crispum*), a medicinal and edible plant of the family Apiaceae, originated in Greece at the end of the 3rd century BC [1]. In addition to its original decorative and spice uses for thousands of years [2], parsley has received increasing attention as a potential functional food because of its therapeutic properties, including analgesic, immunomodulatory, antioxidant, cardiovascular, and antimicrobial activities [3]. Parsley is rich in specialized metabolites with medicinal properties [4]. Parsley extracts contain various metabolites, including flavonoids (apigenin, luteolin, chrysoeriol, and quercetin) [5,6], carbohydrates (apiose), oils (myristicin and apiol) [7,8], and coumarins. Of these, apiin (apigenin-7-*O*-β-[D-apiofuranosyl-(1→2)-β-D-glucopyranoside]) is a major flavonoid glycoside in parsley (up to 3.7 g/100 g dry weight) [9]. Parsley produces some flavonoid apiosylglucosides, as well as apiin. Apiin is resistant to ultraviolet irradiation [10] and has antioxidant properties [11]. It also acts in humans, has a potential role in alleviating hyperuricemia [12], has potential inhibitory effects on SARS-CoV-2 [13,14], and has antipsoriatic potential [15].

The first step in studying the structure–function relationship of flavonoid glycosides, including apiin, is to understand the biosynthetic pathway and identify the genes encoding their biosynthetic enzymes. Extensive studies have shown that the genes encoding flavonoid biosynthetic enzymes are conserved among different plants to some extent [16]. Apigenin, an aglycon of apiin, and biosynthetic enzymes, including phenylalanine ammonium lyase, cinnamate-4-hydroxylase, 4-coumarate:CoA ligase, chalcone synthase, chalcone isomerase, and flavone synthase I, have been identified in plants other than parsley (Figure 1) [17]. The metabolons (multiple enzyme complexes) formed by these enzymes are thought to efficiently biosynthesize flavonoids [18]. Enzymes belonging to glycosyltransferase family 1 (GT1) are responsible for the addition of sugars to flavonoids, resulting in the structural diversity of flavonoids [19,20]. GT1 comprises an extensive array of UDP-sugar glycosyltransferases (UGTs) [21]. UGTs transfer sugar residues from nucleotide sugars to flavonoids, thereby imparting distinct physicochemical properties, such as stability, water solubility, and bioactivity [22], making them one of the targets of drug exploration [23,24]. Some UGTs are related to cold resistance [25] and developmental responses [26]. The parsley UGTs for apiin biosynthesis, which have not yet been identified, are also essential in studies on the functional significance of apiin.

Apiin is produced by the Apiaceae family, including parsley and celery, as well as by the Asteraceae, Solanaceae, and Fabaceae families [27,28]. Of these, parsley is known to produce up to 3.7 g per 100 g dry weight [9]. Therefore, it would be useful to identify the glycosyltransferase genes involved in apiin biosynthesis in parsley to analyze the function of apiin and to clarify the mechanism by which apiin is produced in large amounts. Two glycosyltransferases, apigenin:7-*O*-β-glucosyltransferase (GlcT) and apigenin 7-*O*-β-glucoside:β1-2 apiosyltransferase (ApiT), are necessary for apiin biosynthesis (Figure 1). This GlcT is an inverting glycosyltransferase that transfers glucose from UDP-α-glucose to the 7-*O*-position of apigenin via a β-linkage [29]. This enzyme gene has been identified in several plant species and they are classified into the UGT 71, 73, and 88 groups of GT1 [30,31,32,33,34,35,36,37,38]. Additionally, specific genes from UGT 72 and 75 have also been identified as GlcTs [35,37,38]. These include the plant secondary product glycosyltransferase (PSPG) motif, which consists of 44 amino acid residues conserved near the *C*-terminus with a specific sequence and the UDP-sugar-binding site of the enzyme [39]. Parsley (*Petroselinum crispum*) apigenin:7-*O*-β-glucosyltransferase (PcGlcT) has yet to be identified. It is difficult to identify PcGlcT from the amino acid sequence alone because there are many UGTs with PSPG motifs in the parsley genome [40].

The ApiT is also an inverting glycosyltransferase that transfers apiose residues from UDP-α-apiose to apigenin 7-*O*-glucoside via a β1,2-linkage [41,42,43]. ApiT is one of the glycoside-specific glycosyltransferases (GGTs) that transfer sugars to glycosides. The GGTs have been classified as UGT79, 91, and 94 [40]. The gene encoding ApiT has recently been identified in celery and licorice as *AgApiT* (UGT94AX1) and *GuApiGT* (UGT79B74), respectively [44,45]. These ApiTs contain specific amino acid residues for recognizing apiose residues in UDP-apiose. However, the parsley apigenin 7-*O*-β-glucoside:β1-2 apiosyltransferase (*PcApiT*) gene has not yet been identified.

The aim of this study was to identify the *PcGlcT* and *PcApiT* genes that orchestrate successive glycosylation steps in apiin biosynthesis in parsley. *PcGlcT* and *PcApiT* candidate genes were selected from the parsley RNA-sequencing (RNA-Seq) dataset. Proteins encoded by the most likely genes were analyzed for their biochemical characteristics, including enzymatic activity, substrate specificity, and enzymatic kinetics. The relationship between the expression levels of *PcGlcT* and *PcApiT* and the accumulation of flavonoid glycosides in diverse parsley organs was examined to verify their involvement in apiin biosynthesis.

## 2. Results and Discussion

### 2.1. Gene Identification of PcGlcT and PcApiT

To identify the *PcGlcT* and *PcApiT* genes, candidate genes were screened from the transcriptome data using RNA-Seq of young parsley true leaves, where a substantial amount of apiin is biosynthesized [9]. These transcriptomes were assembled de novo into a total of 66,195 genes and 112,154 transcripts, and their expression levels were quantified in terms of Transcripts Per Million (TPM). PcGlcT is a 7-*O*-UGT that transfers glucose residues from UDP-glucose to the 7-*O*-position of flavonoids. The 7-*O*-UGTs have been found in UGT71, 72, 73, 75, and 88. In the catalog of parsley transcripts, ten genes were identified and selected as candidate genes for *PcGlcT* (Figure 2 and Table 1). The gene, *Ubk32_id_1456*, with highest TPM value was designated as *PcGlcT*.

The PcApiT candidates were recognized as GGT, consisting of UGT79, 91, and 94 groups belonging to Orthogroup 8 (OG8) [40]. In the parsley transcript catalog, 10 potential *PcApiT* candidates were selected (Figure 3 and Table 2). Of these candidates, the gene Ubk32_id_4398, which had the highest TPM, was tentatively designated as *PcApiT*.

Recombinant PcGlcT was heterologously expressed in *Escherichia coli* as a protein with a molecular mass of 53 kDa (Appendix A). Enzyme activity assays were conducted using apigenin as the acceptor substrate and UDP-Glc as the donor substrate. The enzyme product, apigenin 7-*O*-glucoside, appearing at a retention time of 10.3 min (Figure 4A,B) was quantitatively detected. Thus, PcGlcT, registered as UGT88Z2, was biochemically identified as a glucosyltransferase involved in apiin biosynthesis in parsley plants. The amino acid sequence alignment of PcGlcT and the known apigenin:7-*O*-β-glucosyltransferases revealed that PcGlcT contained both the PSPG motif and the GSS motif, both of which are shared among glucosyltransferases (Appendix A).

PcApiT was heterologously expressed in *E. coli* as a protein with a molecular mass of 49 kDa (Appendix A). Incubation of PcApiT with apigenin 7-*O*-glucoside and UDP-Api produced apiin as an enzyme product with a retention time of 11.1 min (Figure 4A,C). Thus, PcApiT was confirmed to be a parsley apigenin 7-*O*-glucoside apiosyltransferase and registered as UGT94AX2. The amino acid sequence of PcApiT (UGT94AX2) is most similar (78%) to that of AgApiT (UGT94AX1) [44] (Appendix A). PcApiT is the only parsley GGT conserved with the Ile139, Phe140, and Leu356 residues in AgApiT, which is important for the recognition of apiose residues in donor substrates [44]. Recently, GuApiGT (UGT79B74), an apiosyltransferase involved in the biosynthesis of flavonoid apiosylglucosides in the Leguminosae plant, *Glycyrrhiza uralensis*, was identified [45]. PcApiT shares 20% amino acid sequence homology with GuApiGT. The amino acid residues recognized in the apiose portion of UDP-Api in GuApiGT [45] differed from those in PcApiT and AgApiT [44]. This suggested that PcApiT and AgApiT evolved from the same ancestor, although different from the ancestor of GuApiGT.

### 2.2. Biochemical Characterization of PcGlcT and PcApiT

The optimum pH for PcGlcT activity was approximately 9 and the enzyme remained active at pH 7–10 (Figure 5A). Its optimum temperature under the reaction conditions used in this study was ~25 °C (Figure 5B). Even at temperatures below 10 °C, the activity was half that at 25 °C, and above 40 °C, there was little activity. The optimum pH for PcApiT was approximately 7, and it remained active at pH 5–8 (Figure 5C). PcApiT exhibited an optimum temperature at 25 °C, with low activity below 10 °C and above 35 °C (Figure 5D). Some apigenin 7-*O*-glucoside-synthetic glucosyltransferases from other plants have an optimum pH of approximately 9, which is similar to that of PcGlcT [33,37]. Other glucosyltransferases display an optimum pH of 7 [46,47,48]. The relationship between optimum pH values and the structure of glucosyltransferases is currently unknown. Most UGTs, including PcGlcT and PcApiT, are thought to be expressed in the cytoplasm, and these enzymes are thought to function at ~pH 7. The enzymatic activity of PcGlcT at pH 7 was approximately one sixth that at pH 9, and its activity was comparable to that of other glucosyltransferases.

To ascertain acceptor substrate specificity, PcGlcT was allowed to act on a variety of sugar acceptors. It was most active against flavones, such as luteolin and apigenin, followed by quercetin, classified as a flavonol, and naringenin, classified as a flavan (Figure 6A). It also acted on genistein, an isoflavone and chrysoeriol, a flavone. PcGlcT transfers the glucose residue to the hydroxyl group at position 7 in all these compounds. It did not act on quercetin 3-*O*-glucoside, apigenin 7-*O*-glucoside, or apiin. The sugar donor of PcGlcT exhibited a preference for UDP-Glc among the seven sugar nucleotides when apigenin was used as the sugar acceptor (Figure 6B). Thus, PcGlcT exhibits substrate specificity as a glucosyltransferase involved in the biosynthesis of flavone glucosides, including apigenin 7-*O*-β-glucoside.

PcGlcT exhibits *K*_m_ and *k*_cat_ values of 320 ± 70 μM and 0.62 ± 0.05 s^−1^, respectively, for apigenin. Similarly, it exhibits *K*_m_ and *k*_cat_ values of 610 ± 110 μM and 0.62 ± 0.05 s^−1^, respectively, for UDP-Glc (Appendix A). A comparison of its kinetic parameters with those of other apigenin:7-*O*-β-glucosyltransferases showed that the *k*_cat_ value of PcGlcT was higher than that of most other glucosyltransferases, resulting in a higher efficiency in the synthesis of apigenin 7-*O*-glucoside (Table 3).

When PcApiT was allowed to act on several acceptor substrates with UDP-apiose as the donor substrate, it was most active on apigenin 7-*O*-glucoside. PcApiT had the greatest effect on the 7-*O*-glucoside forms of flavones, including apigenin, chrysoeriol, and luteolin, followed by the 7-*O*-glucosides of naringenin (flavan) and quercetin (flavonol) (Figure 6C). No activity was observed toward quercetin 3-*O*-glucoside, apigenin, or apiin. Of the seven nucleotide sugars, PcApiT exclusively utilized UDP-Api as a donor substrate when apigenin 7-*O*-glucoside was used as the acceptor substrate (Figure 6D). The substrate specificities of PcApiT are similar to those of AgApiT from celery [44], and they were shown to be involved in the biosynthesis of chrysoeriol and luteolin 7-*O*-apiosylglucosides produced in parsley, in addition to apiin.

The *K*_m_ and *k*_cat_ values of PcApiT for apigenin 7-*O*-glucoside were 81 ± 20 μM and (3.2 ± 0.3) × 10^−3^ s^−1^, respectively. The *K*_m_ and *k*_cat_ values for UDP-Api were 360 ± 40 μM and (4.5 ± 0.2) × 10^−3^ s^−1^, respectively (Appendix A and Table 4). PcApiT exhibited higher *K*_m_ and *k*_cat_ values, resulting in *k*_cat_/*K*_m_ values similar to those of AgApiT [44].

### 2.3. PcGlcT and PcApiT Expression and Flavonoid Glucosides Accumulation Profiles in Parsley

The correlation between the expression levels of glycosyltransferases and the accumulation of flavonoid glucosides in different organs of parsley was investigated. Parsley flavone synthase I (PcFNSI), [49] which catalyzes the conversion of naringenin to apigenin, was used as the reference. The real-time quantitative reverse transcription polymerase chain reaction (qRT-PCR) was performed for *FNSI*, *PcGlcT*, and *PcApiT* across diverse organs and leaves in the developmental stages of parsley. The expression levels of *PcGlcT* and *PcApiT* were the highest in true leaves but low in seeds, roots, and stems (Figure 7A,B). In the true-leaf developmental stages, their expression was high at the 0.5–1.0 cm stage and decreased gradually as the developmental stages progressed. Their expression profiles were similar to those of *PcFNSI* (Figure 7C).

The contents of flavonoid 7-*O*-apiosylglucosides (total of apiin, chrysoeriol 7-*O*-apiosylglucoside, and luteolin 7-*O*-apiosylglucoside) across various parsley organs and true-leaf developmental stages were determined. The flavonoid 7-*O*-apiosylglucosides were the highest in the seeds and true leaves (Figure 7D); of the developmental stages in the true leaves, it was highest at the 0.5–1.0 cm stage and decreased gradually at later stages (Figure 7D). This profile is correlated with the expression of *PcGlcT*, *PcApiT*, and *PcFNSI* in roots, stems, and leaves. This reaffirms the pivotal role of PcGlcT and PcApiT in the biosynthesis of flavonoid 7-*O*-apiosylglucosides, including apiin, in parsley. The flavonoid 7-*O*-apiosylglucoside content in the seeds was high and inconsistent with the expression levels of the biosynthetic genes. This was thought to reflect the accumulation of flavonoid 7-*O*-apiosylglucosides biosynthesized during seed development.

*PcGlcT*, *PcApiT*, and *PcFNSI* have been identified as apiin-biosynthetic enzyme genes in parsley. It is necessary to identify other enzyme genes involved in apiin biosynthesis in parsley to analyze the function of apiin or apiose residues and the mechanism of apiin biosynthesis. It is possible to produce apiin via fermentation using the enzyme genes identified in this study. This is the first step in the biological research on apiin.

## 3. Materials and Methods

### 3.1. Plant Cultivation

The parsley seeds (cultivar ‘Paramount’) were obtained from Takii & Co. Ltd. (Kyoto, Japan). The seeds were sown and grown on a 1:1 soil mixture of Metro-Mix (Sun Gro Horticulture, Agawam, OH, USA) and vermiculite (1:1) (pH 6.0–6.5) at 22 °C under a photoperiod of 16 h light and 8 h darkness, with a light intensity of 132 μmol⋅m^−2^⋅s^−1^.

### 3.2. PcGlcT and PcApiT Candidate Genes

Total RNA was isolated from the true leaves of parsley using an RNeasy Plus Mini Kit (Qiagen, Hilden, Germany) according to the manufacturer’s instructions. RNA-seq (Sequence Read Archive ID: DRR505880) was performed using GENEWIZ (Tokyo, Japan). The library was sequenced using a DNBSEQ-G400 sequencer (MGI) with 150 bp paired-end reads. The reads were trimmed using Trimmomatic version 0.39, followed by de novo assembly using Trinity version 2.8.5, with a k-mer size of 32. Amino acid sequences were predicted using TransDecoder version 5.5.0, based on the Trinity results. Expression levels were quantified using Salmon version 0.14.1. Candidate *PcGlcT* and *PcApiT* genes were selected using BLAST 2.12.0+ with known sequences of 7-*O*-UGTs in UGT71, 72, 73, 75, and 88 for *PcGlcT*, and UGT79, 91, and 94 for *PcApiT* as queries within the transcript catalog created via de novo assembly. A phylogenetic tree was constructed using TBtools based on the maximum-likelihood algorithm (IQ-TREE) and a bootstrap test with 1000 replicates [50,51]. The web application, tvBOT, was used to visualize, modify, and annotate the tree [52]. Multiple sequence alignments were conducted on the ClustalOmega website and [53] edited using Jalview [54].

### 3.3. Heterologous Expression of PcGlcT and PcApiT

The coding regions of *PcGlcT* and *PcApiT* were chemically synthesized as codon-optimized genes (Eurofins Genomics, Tokyo, Japan) for subsequent protein expression in *Escherichia coli* cells. The DNA sequences of *PcGlcT* (UGT88Z2) and *PcApiT* (*UGT94AX2*) have been deposited in DDBJ/ENA/GenBank under the accession numbers LC782275 and LC782274, respectively. *PcGlcT* was amplified using the pET28b_PcGlcT_F and pET28b_PcGlcT_R primers and cloned into the NdeI/XhoI sites of the pET28b vector (Takara Bio, Kusatsu, Japan). Similarly, *PcApiT* was amplified using the pColdProS2_PcApiT_F and pColdProS2_PcApiT_R primers and subsequently inserted into the NdeI/XbaI sites of the pColdProS2 vector (Takara Bio) (Appendix A).

*Escherichia coli* BL21 (DE3) cells transformed with the resulting vectors were cultured in LB medium supplemented with 100 mg/mL ampicillin at 37 °C until reaching an optical density read at 600 nm (OD_600_) of 0.6. Isopropyl β-thiogalactopyranoside was then added to the culture at a final concentration of 1 mM, and the incubation was continued at 15 °C for 24 h. Then, the cells were harvested via centrifugation at 5000× *g* at 4 °C for 5 min. Cell lysis was performed using BugBuster Protein Extraction Reagent (Millipore, Burlington, MA, USA) supplemented with 10 mM sodium phosphate buffer (pH 7.4), 5 U/mL benzonase, and 1 kU/mL lysozyme, and incubated for 20 min. The cellular debris was removed via centrifugation at 20,000× *g* at 4 °C for 20 min. The mixture was then filtered through a 0.45 μm filter, and the filtrate was applied to a 1 mL HisTrap HP column (GE Healthcare Life Sciences, Chicago, IL, USA) pre-equilibrated with buffer 1 (50 mM sodium phosphate buffer, 500 mM sodium chloride, and 20 mM imidazole at pH 7.4). Both PcGlcT and PcApiT proteins were eluted with elution buffer (50 mM sodium phosphate buffer, 500 mM sodium chloride, and 300 mM imidazole at pH 7.4). The ProS2_PcApiT fusion protein underwent digestion with HRV3C protease at 4 °C for 12 h to remove the proS2 tag. The resulting PcApiT protein was purified as the flow-through fraction on a 1 mL HisTrap HP column equilibrated with buffer 1. The purified protein was concentrated via ultrafiltration using an Amicon Ultra-0.5 mL device with a 10 kDa cutoff at 14,000× *g* at 4 °C for several cycles. The proteins were subsequently subjected to sodium dodecyl sulphate–polyacrylamide gel electrophoresis (SDS-PAGE) and visualized using Coomassie Blue staining.

### 3.4. Enzymatic Assays of PcGlcT and PcApiT

To assess the catalytic activity of the purified recombinant PcGlcT across various substrates, a reaction mixture consisting of 1 mM sugar donors, 1 mM sugar acceptors, 2.0–5.0 μg/μL of PcGlcT, and 100 mM Tris-HCl buffer (pH 9.0) containing 50 mM NaCl was incubated at 23 °C for 30 min. The range of sugar acceptors included apigenin (Cayman Chemical, Ann Arbor, MI, USA), apigenin 7-*O*-β-D-glucoside, apiin (Ark Pharm, Arlington Heights, IL, USA), naringenin, luteolin, quercetin, genistein (TCI Chemicals, Toyo, Japan), and chrysoeriol (Extrasynthese, Genay Cedex, France). The sugar donors included UDP-Glc, UDP-Gal, UDP-GlcNAc (Fujifilm Wako Chemical Corporation, Osaka, Japan), UDP-GlcA, GDP-Fuc (Merck, Darmstadt, Germany), and UDP-Ara*f* (Peptide Institute, Ibaraki, Japan), whereas UDP-Xyl [55], UDP-Api [43], and UDP-GalA [56] were prepared according to previously described methods.

Similarly, for assessing the catalytic activity of recombinant PcApiT against different substrates, 3.0–5.0 µg/µL of PcApiT was incubated with 100 μM acceptor substrate, 1 mM sugar nucleotide, and 100 mM Tris-HCl buffer (pH 7.0) containing 50 mM NaCl at 23 °C for 1 h. The acceptor substrates included apigenin 7-*O*-β-D-glucoside, apiin, apigenin, chrysoeriol 7-*O*-β-D-glucoside (ChemFaces, Wuhan, China), naringenin 7-*O*-β-D-glucoside, luteolin 7-*O*-β-D-glucoside, quercetin 7-*O*-β-D-glucoside, and quercetin 3-*O*-β-D-glucoside (Extrasynthese).

After the reactions were stopped via incubation at 100 °C for 3 min, the samples were then subjected to reversed-phase high-performance liquid chromatography (HPLC) analysis for quantification of the substrate and product. The *K*_m_ and *k*_cat_ values for PcGlcT and PcApiT were determined using assays with varying substrate concentrations of apigenin (25–1500 μM) and UDP-Glc (50–1500 μM), and apigenin 7-*O*-β-D-glucoside (12.5–2000 μM) and UDP-apiose (15–1000 μM), respectively. The optimal pH was determined across various buffer pHs (100 mM) with 50 mM sodium chloride, including sodium acetate buffer (pH 4.0–6.0), sodium phosphate buffer (pH 6–7.5), HEPES–KOH buffer (pH 7–9.0), Tris-HCl buffer (pH 7–9.5), and glycine–NaOH buffer (pH 10.0–10.5). Likewise, the optimal temperature was assessed across a range of temperatures (10–50 °C) for each reaction mixture.

Reverse-phase HPLC was used to determine the activities of PcGlcT and PcApiT by quantifying the peak areas of the substrates and products. Specifically, 10 µL of the quenched reaction mixture was applied onto an Inertsil ODS-3 column (4.6 × 250 mm, GL Sciences, Tokyo, Japan) at a flow rate of 1.0 mL/min. The chromatographic conditions consisted of an isocratic flow of 25% acetonitrile containing 0.1% trifluoroacetic acid for the initial 5 min, followed by a linear gradient from 25% to 55% acetonitrile over 20 min for PcGlcT. For PcApiT, the conditions involved an initial isocratic flow of 20% acetonitrile containing 0.1% trifluoroacetic acid for 5 min, followed by a linear gradient from 20% to 40% acetonitrile for 20 min. For the detection and quantification of individual compounds, apigenin, chrysoeriol, luteolin, naringenin, and quercetin were detected and quantified based on absorbance at 330, 330, 350, 280, and 254 nm, respectively.

### 3.5. Gene Expression Analysis

The expression patterns of *PcGlcT* and *PcApiT* in different organs and at various developmental stages were analyzed using quantitative reverse transcription-polymerase chain reaction (qRT-PCR). Total RNA was extracted from different parsley tissues, including seeds, roots, stems, and true leaves at different developmental stages (leaf length 0–0.5 cm, 0.5–1.0 cm, 1.0–1.5 cm, 1.5–2.0 cm, and 2.0–2.5 cm), using the RNeasy Mini Kit (Qiagen). Subsequently, total cDNA was synthesized using the PrimeScript II 1st strand cDNA synthesis kit (Takara Bio), following the manufacturer’s protocols. The qRT-PCR reaction mixture (20 μL) included 2.0 µL of cDNA (50 ng/µL), 0.8 µL of each primer (final concentration 0.4 μM), 10 μL of TB Green Premix Ex Taq II (Tli RNaseH Plus), 0.4 μL of ROX reference dye, and 6.0 μL of RNA-free water [57]. The qRT-PCR protocol consisted of a holding stage at 95 °C for 30 s, followed by a cycling stage of 40 cycles at 95 °C for 5 s and 60 °C for 30 s. The melt-curve stage involved incubation at 95 °C for 15 s, 60 °C for 1 min, and 95 °C for 15 s. The primer sequences are listed in Appendix A. The housekeeping gene, *EF-1α*, was selected as the reference gene for normalizing transcript abundances [58]. Relative expression levels were determined using the ΔΔCt method [59], with the leaf sample at 0.5 cm set as the reference (assigned a value of 1). The resulting data represent the average of three replicates.

### 3.6. Quantification of Flavonoid 7-O-β-Apioglucosides in Parsley

Parsley samples (20 mg each of seeds, roots, stems, and leaves at different developmental stages) were pulverized in liquid nitrogen and subsequently extracted with a mixture of methanol and water (1.0 mL, 60:40, *v*/*v*). Extraction was performed using an ultrasonic sonicator operating at 40 kHz and 100 W for 60 min at 25 °C. The resulting extract was then filtered through a 0.45 µm Minisart Syringe Filter (Sartorius, Göttingen, Germany). Subsequently, a 10 µL aliquot of the filtered extract was injected into the HPLC analytical column for subsequent analysis. To detect and quantify flavonoid 7-*O*-β-apioglucosides in parsley, HPLC analysis was conducted using a GL Sciences Inert Sustain C18 column (4.6 × 250 mm, 5 μm) at a flow rate of 1.0 mL/min. The mobile phase consisted of a mixture of solvent A (0.1% formic acid in water) and solvent B (0.1% formic acid in acetonitrile). The gradient profile included a linear increase from 10% to 26% B (*v/v*) over 40 min, followed by a gradual increase to 65% B at 70 min and a final increase to 100% B at 71 min, maintained until 75 min. Absorbance was monitored at 350 nm to determine peak intensities [44,60].

## Figures and Tables

**Figure 1 ijms-24-17118-f001:**
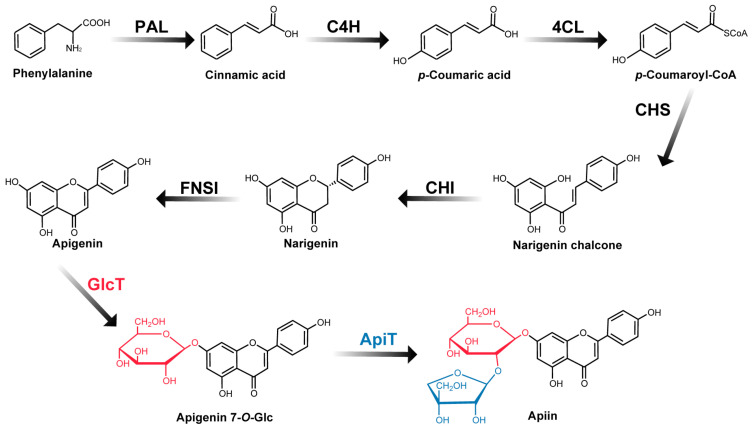
Biosynthesis of apiin. PAL, phenylalanine ammonia lyase; C4H, cinnamate 4-hydroxylase; 4CL, 4-coumarate CoA ligase; CHS, chalcone synthase; CHI, chalcone isomerase; FNSI, flavone synthase I; GlcT, glucosyltransferase; ApiT, apiosyltransferase.

**Figure 2 ijms-24-17118-f002:**
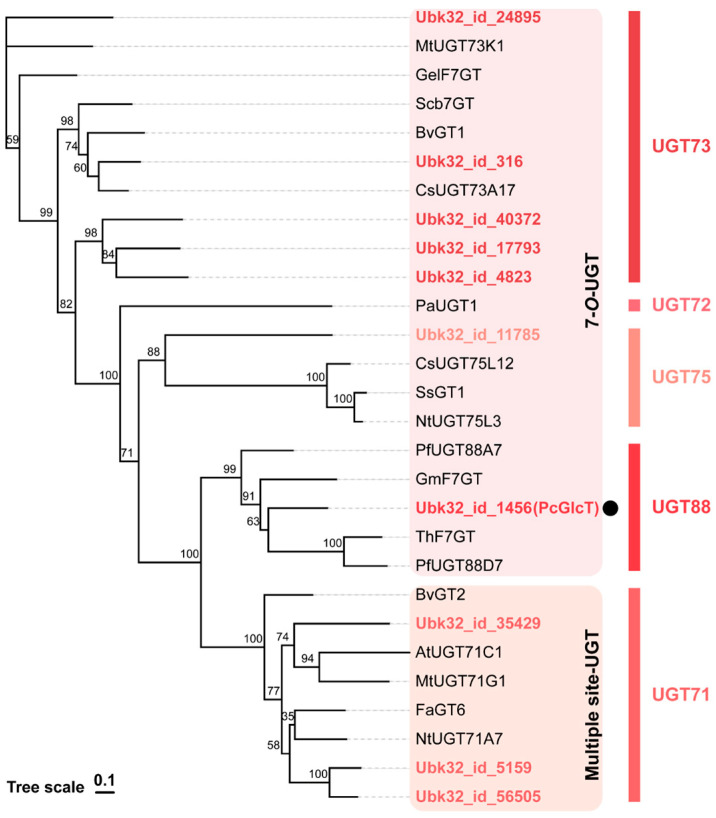
Phylogenetic analysis of *PcGlcT* candidate genes with known related glucosyltransferases shows activity specific toward flavonoid 7-OH or multiple OH positions in UGT 71, 72, 73, 75, and 88. PcGlcT is marked with a black circle.

**Figure 3 ijms-24-17118-f003:**
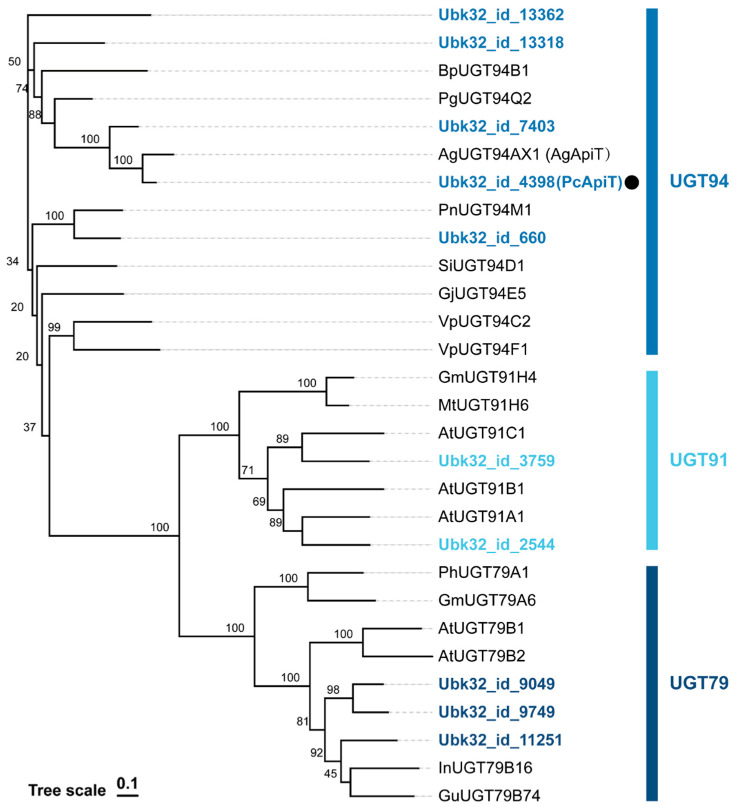
Phylogenetic analysis of *PcApiT* candidate genes and known GGTs in UGT79, 91, and 94. PcApiT, marked with a black circle, is classified as UGT 94.

**Figure 4 ijms-24-17118-f004:**
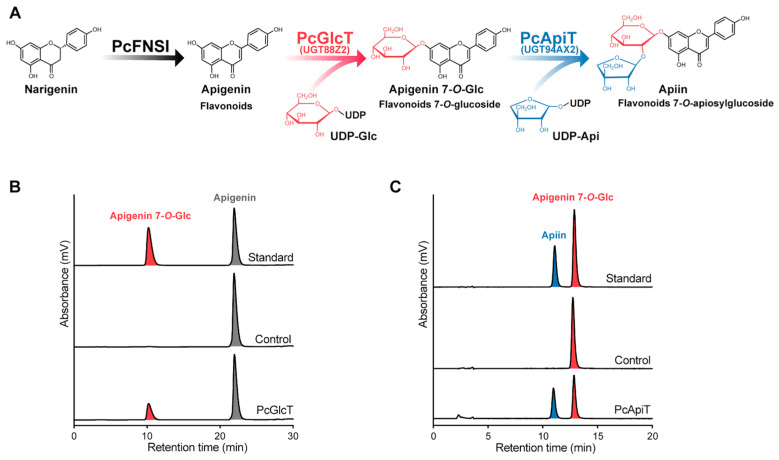
Enzyme activities of PcGlcT and PcApiT. (**A**) The sequential glycosylation process of apiin biosynthesis in parsley. PcGlcT catalyzes apigenin with UDP-Glc to produce apigenin 7-*O*-glucoside, whereas PcApiT exhibits catalytic activity toward apigenin 7-*O*-glucoside with UDP-Api to produce apiin. (**B**) HPLC analysis of the enzymatic reactions of PcGlcT compared with the authentic standard (apigenin 7-*O*-glucoside). (**C**) HPLC analysis of the enzymatic reactions of PcApiT compared with the authentic standard (apiin).

**Figure 5 ijms-24-17118-f005:**
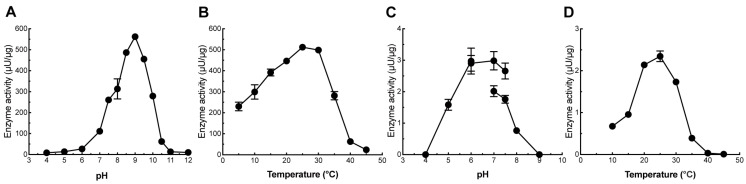
pH and temperature dependencies of PcGlcT and PcApiT. (**A**) Optimal reaction pH for the PcGlcT enzyme. (**B**) Optimal reaction temperature for the PcGlcT enzyme. (**C**) Optimal reaction pH for the PcApiT enzyme. (**D**) Optimal reaction temperature for the PcApiT enzyme. For PcGlcT, UDP-Glc and apigenin were used as substrates. For PcApiT, UDP-Api and apigenin 7-*O*-glucoside were used as substrates.

**Figure 6 ijms-24-17118-f006:**
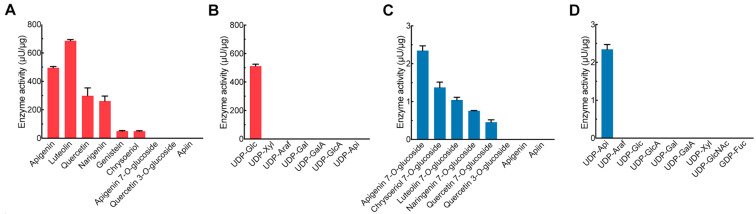
Substrate specificity of recombinant PcGlcT and PcApiT. (**A**) Acceptor substrate specificity of PcGlcT. UDP-Glc was used as donor substrate. (**B**) Donor substrate specificity of PcGlcT. Apigenin was used as acceptor substrate. (**C**) Acceptor substrate specificity of PcApiT. UDP-Api was used as donor substrate. (**D**) Donor substrate specificity of PcApiT. Apigenin 7-*O*-glucoside was used as acceptor substrate. All data are presented as mean values with standard errors from three replicates.

**Figure 7 ijms-24-17118-f007:**
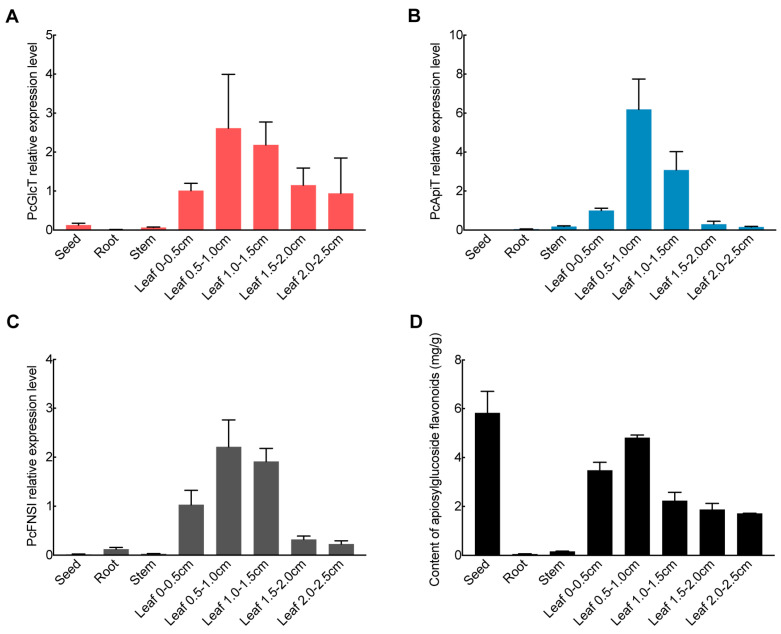
*PcGlcT* and *PcApiT* gene expression profiles and flavonoid 7-*O*-β-apiosylglucoside contents in parsley. Expression profile of (**A**) *PcGlcT*, (**B**) *PcApiT*, and (**C**) *PcFNSI* in different organs and true-leaf developmental stages using qRT-PCR. (**D**) Content of flavonoid 7-*O*-β-apiosylglucosides in different organs and true leaves developmental stages. Each bar represents mean values and standard deviations from three biological replicates.

**Table 1 ijms-24-17118-t001:** PcGlcT candidates from transcriptome catalog of parsley RNA-Seq data.

Protein ID	UGT Family	Regioselectivity	TPM
Ubk32_id_1456 (PcGlcT)	UGT88	7-*O*-UGT	59.7
Ubk32_id_316	UGT73	7-*O*-UGT	16.1
Ubk32_id_5159	UGT71	Multiple site-UGT	9.8
Ubk32_id_11785	UGT75	7-*O*-UGT	9.1
Ubk32_id_35429	UGT71	Multiple site-UGT	4.2
Ubk32_id_4823	UGT73	7-*O*-UGT	1.8
Ubk32_id_17793	UGT73	7-*O*-UGT	1.6
Ubk32_id_24895	UGT73	7-*O*-UGT	1.4
Ubk32_id_40372	UGT73	7-*O*-UGT	1.1
Ubk32_id_56505	UGT71	Multiple site-UGT	1.0

The PcGlcT candidates were selected from the transcriptome catalog of parsley. Red is the most probable candidate protein of PcGlcT.

**Table 2 ijms-24-17118-t002:** PcApiT candidates from transcriptome catalog of parsley RNA-Seq data.

Protein ID	UGT Family	TPM	Identity to AgApiT (%)
Ubk32_id_4398	UGT94	147.1	78
Ubk32_id_2544	UGT91	61.4	28
Ubk32_id_11251	UGT79	25.8	26
Ubk32_id_9049	UGT79	6.7	27
Ubk32_id_660	UGT94	4.9	41
Ubk32_id_9749	UGT79	3.5	24
Ubk32_id_7403	UGT94	3.1	62
Ubk32_id_13362	UGT94	2.1	39
Ubk32_id_3759	UGT91	1.6	30
Ubk32_id_13318	UGT94	1.5	44

The amino acid sequences of parsley UGTs from the UGT 79, 91, and 94 groups were selected based on the RNA-Seq database. Blue is the most probable candidate of PcApiT.

**Table 3 ijms-24-17118-t003:** Kinetic parameters of PcGlcT and other apigenin:7-*O*-glucosyltransferases from different plant species.

Source	Enzyme	Substrate	*K*_m_ (μM)	*k*_cat_ (s^−1^)	k_cat_/K_m_ (s^−1^/mM)
Parsley	PcGlcT	Apigenin	320 ± 70	(6.2 ± 0.5) × 10^−1^	1.97 ± 0.69
Lamiales	UGT88D7 [30]	Apigenin	8.5 ± 1.9	(6.0 ± 0.3) ×10^−3^	0.71
Liverwort	PaUGT1 [35]	Apigenin	22.2 ± 3.3	(2.2 ± 0.1) ×10^−3^	0.098
Tea	UGT73A17 [33]	Apigenin	5.5 ± 1.5	2.3 × 10^−3^	0.54
Tea	UGT75L12 [37]	Apigenin	4.6 ± 0.2	2.1 × 10^−2^	4.54
Ginkgo biloba	UGT716A1 [38]	Apigenin	230 ± 0.0	1.7 × 10^−2^	0.073
Parsley	PcGlcT	UDP-Glc	610 ± 110	(6.2 ± 0.5) × 10^−1^	1.01 ± 0.44
Lamiales	UGT88D7 [30]	UDP-Glc	540 ± 20		0.011
Tea	UGT75L12 [37]	UDP-Glc	233.3 ± 41.7	14.9 × 10^−2^	0.64

Data are presented as mean ± SD (*n* = 3).

**Table 4 ijms-24-17118-t004:** Kinetic parameters of PcApiT and AgApiT.

Source	Enzyme	Substrate	*K*_m_ (μM)	*k*_cat_ (s^−1^)	k_cat_/K_m_ (s^−1^/mM)
Parsley	PcApiT	Apigenin 7-*O*-Glc	81 ± 20	3.2 ± 0.3	40 ± 17
Celery	AgApiT [44]	Apigenin 7-*O*-Glc	15 ± 3	0.88 ± 0.05	58 ± 15
Parsley	PcApiT	UDP-Api	360 ± 40	4.5 ± 0.2	12 ± 5
Celery	AgApiT [44]	UDP-Api	8.6 ± 0.6	0.65 ± 0.01	76 ± 6

Data are presented as mean ± SD (*n* = 3).

## Data Availability

Data is contained within the article and Appendix A.

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
