# Peer review of "Biochemical Characterization of Parsley Glycosyltransferases Involved in the Biosynthesis of a Flavonoid Glycoside, Apiin"

_ijms, 2023, doi:10.3390/ijms242317118_

Round 1

Reviewer 1 Report

Comments and Suggestions for Authors

This manuscript describes identification and characterisation of two glycosyltransferases involved in biosynthesis of apiin. The authors first identified genes of plausible glycosyltransferases in parsley and then recombinantly expressed and tested such enzymes for the conversion of apigenin and apigenin-7-O-GlcNac to corresponding products. The manuscript is well written. This work is of interest to enzymologists and protein biochemists. The manuscript will be suitable for publication once the following minor points are addressed.

Based on the results in Figure 3, it seems that only a small portion of the substrate was converted into product. Additional experiments need to be carried out to explore the possibility to obtain both products in higher yields, quantitatively if possible. This might require an increased amount of enzyme or longer time.

The cascade reaction needs to be carried out. Is apiin produced when apigenin, PcGlcT and PcApiT are incubated together? As both enzymes are active at 25°C, this should be feasible to explore at pH 7.5 or 8, under conditions that keep both enzymes active.

The authors need to rationalise why both enzymes display increased specificity for UDP-sugar cosubstrate over apigenin-type substrate, as shown in Figure 5. Why changes in the cosubstate are not tolerated, whereas in the substrate they are tolerated? A biostructural support from related enzymes could be helpful in this regard.

The SI is too basic and should contain more information. SDS-PAGE data for the recombinantly produced PcGlcT and PcApiT should be shown. The exact sequences of both enzymes also need to be provided.

Author Response

This manuscript describes identification and characterisation of two glycosyltransferases involved in biosynthesis of apiin. The authors first identified genes of plausible glycosyltransferases in parsley and then recombinantly expressed and tested such enzymes for the conversion of apigenin and apigenin-7-O-GlcNac to corresponding products. The manuscript is well written. This work is of interest to enzymologists and protein biochemists. The manuscript will be suitable for publication once the following minor points are addressed.

Thank you very much for taking the time to evaluate our manuscript accurately. All your suggestions are greatly appreciated as they give us an opportunity to reevaluate the content of this manuscript. Below are our sincere responses to each of this reviewer’s comments.

  1. Based on the results in Figure 3, it seems that only a small portion of the substrate was converted into product. Additional experiments need to be carried out to explore the possibility to obtain both products in higher yields, quantitatively if possible. This might require an increased amount of enzyme or longer time.

To quantitatively evaluate enzymes, it is common to measure the initial velocity of the enzyme. Therefore, quantitative evaluation of the enzyme was performed under reaction conditions where the amount of enzyme product was low (Figure 4B). Considering the purpose of this study (characterization of two glycosyltransferases), we believe that experiments that increase the amount of enzyme or lengthen the reaction time are unnecessary.

  1. The cascade reaction needs to be carried out. Is apiin produced when apigenin, PcGlcT and PcApiT are incubated together? As both enzymes are active at 25°C, this should be feasible to explore at pH 7.5 or 8, under conditions that keep both enzymes active.

Because the main purpose of this paper is to characterize each of the two glycosyltransferases, the results of the coexistence of the two enzymes to confirm the production of apiin are excluded. We are attempting to produce apiin in Escherichia coli by several enzymes, including these two enzymes. This request is beyond the purpose of this paper and will be presented in the next paper. We would appreciate your understanding of this situation.

  1. The authors need to rationalise why both enzymes display increased specificity for UDP-sugar cosubstrate over apigenin-type substrate, as shown in Figure 5. Why changes in the cosubstate are not tolerated, whereas in the substrate they are tolerated? A biostructural support from related enzymes could be helpful in this regard.

The difference in recognising these glycosyltransferases for donor and acceptor substrates is of interest from structural biology. As you pointed out, this study revealed that the specificities for their donor substrate are strict, but their specificities for the acceptor substrate are relatively loose. This is one of the findings of this study. Structural biological studies to explain these results are so important that we would like to address them as our next research target. However, because this is beyond the scope of the present paper, we ask that you permit us to publish it in its present form.

  1. The SI is too basic and should contain more information. SDS-PAGE data for the recombinantly produced PcGlcT and PcApiT should be shown. The exact sequences of both enzymes also need to be provided.

We have already shown SDS-PAGE data for the recombinant PcGlcT and PcApiT in Supplementary Figure 2. We added the SDS-PAGE data for the recombinant PcApiT with a ProS2tag. The DDBJ/Genbank registry numbers are shown so that the exact sequences of the two glycosyltransferases can be obtained (Line 274-276). The sequences are also shown in Supplementary Figure S1.

Reviewer 2 Report

Comments and Suggestions for Authors

The authors of the manuscript "Biochemical Characterization of Parsley Glycosyltransferases Involved in the Biosynthesis of a Flavonoid Glycoside, Apiin" attempted to identify the PcGlcT and PcApiT genes, which organize subsequent steps of glycosylation in the biosynthesis of apiin in parsley. The research material was well prepared and methodically carried out correctly. Minor corrections and additions are required in the submitted text. The first suggestion concerns a procedure to improve the reading comprehension process, especially for people who are poorly familiar with research applications. I suggest that before the introduction you include a list of abbreviations and their meanings. The second remark is related to the specification of the soil parameter with the pH indicator, because the synthesis of biologically active compounds depends on it. The third remark is related to the explanation of how the tested compounds were identified using the HPLC method - the text states that they were based on absorbance. For correct identification, you cannot rely only on absorbance, you also need to use standards or a mass spectrometer.

No clearly redacted conclusions.

Author Response

The authors of the manuscript "Biochemical Characterization of Parsley Glycosyltransferases Involved in the Biosynthesis of a Flavonoid Glycoside, Apiin" attempted to identify the PcGlcT and PcApiT genes, which organize subsequent steps of glycosylation in the biosynthesis of apiin in parsley. The research material was well prepared and methodically carried out correctly. Minor corrections and additions are required in the submitted text.

Thank you for taking the time to review our manuscript. The reviewer’s suggestions will help us to better improve this manuscript. We will carefully respond to each comment below.

  1. The first suggestion concerns a procedure to improve the reading comprehension process, especially for people who are poorly familiar with research applications. I suggest that before the introduction you include a list of abbreviations and their meanings.

Thank you for pointing this out to us. We have already mentioned the formal names of the main abbreviations in the text. Take the purpose of this comment, we have added a new figure (Figure 1) of the apiin biosynthesis scheme. We believe that those who are unfamiliar with this field will find it helpful in understanding the main parts of this study.

  1. The second remark is related to the specification of the soil parameter with the pH indicator, because the synthesis of biologically active compounds depends on it.

Thank you for pointing this out. We agreed to follow this opinion. I have recorded the pH of the soil used (Line 253).

  1. The third remark is related to the explanation of how the tested compounds were identified using the HPLC method - the text states that they were based on absorbance. For correct identification, you cannot rely only on absorbance, you also need to use standards or a mass spectrometer.

Thank you for your comment. UV absorbance was used to quantify each compound. The identification of each compound is carried out by comparison with the retention time of the authentic standards on reversed-phase chromatography (upper chromatogram in Figure 4B). This method is a common method for identifying compounds and has been used in similar papers in the past. The compounds other than apigenin-related substances were also identified by the retention time of authentic standards on HPLC.

  1. No clearly redacted conclusions.

The conclusions are summarized in the abstract. We believe this is sufficient because the discussion is not complex in this manuscript.